# Evaluation of Cardiac Substructures Exposure of DIBH-3DCRT, FB-HT, and FB-3DCRT in Hypofractionated Radiotherapy for Left-Sided Breast Cancer after Breast-Conserving Surgery: An In Silico Planning Study

**DOI:** 10.3390/cancers15133406

**Published:** 2023-06-29

**Authors:** Jordan Eber, Martin Schmitt, Nicolas Dehaynin, Clara Le Fèvre, Delphine Antoni, Georges Noël

**Affiliations:** 1Department of Radiation Oncology, Institut de Cancérologie Strasbourg Europe (ICANS), 67033 Strasbourg, France; ma.schmitt@icans.eu (M.S.);; 2Centre Paul Strauss, Strasbourg University, CNRS, IPHC UMR 7178, UNICANCER, 67000 Strasbourg, France

**Keywords:** breast cancer, radiotherapy, cardiac substructures, deep-inspiration breath-hold, helical tomotherapy

## Abstract

**Simple Summary:**

Radiotherapy for left-sided breast cancer (LSBC) can lead to cardiovascular complications. Cardiac-sparing approaches such as deep-inspiration breath-hold (DIBH) and intensity-modulated radiotherapy (IMRT) have been developed to reduce heart exposure. In this in silico study, the cardiac substructure exposure of DIBH three-dimensional conformal radiation therapy (3DCRT), free-breathing (FB)-3DCRT, and FB helical tomotherapy (HT) were compared. Ten patients with left-sided breast cancer who underwent hypofractionated DIBH-3DCRT were selected. DIBH-3DCRT and FB-HT demonstrated a significant dose reduction in the heart, specifically the left anterior descending coronary artery and ventricle, when compared to FB-3DCRT. DIBH-3DCRT was found to be a highly effective cardiac-sparing technique, but in some cases, it did not offer sufficient protection. FB-HT may be an alternative for patients with cardiovascular risks who cannot perform DIBH. These findings suggest that precise delivery of radiation therapy is crucial in reducing cardiac exposure for patients undergoing LSBC radiotherapy.

**Abstract:**

Left-sided breast cancer radiotherapy can lead to late cardiovascular complications, including ischemic events. To mitigate these risks, cardiac-sparing techniques such as deep-inspiration breath-hold (DIBH) and intensity-modulated radiotherapy (IMRT) have been developed. However, recent studies have shown that mean heart dose is not a sufficient dosimetric parameter for assessing cardiac exposure. In this study, we aimed to compare the radiation exposure to cardiac substructures for ten patients who underwent hypofractionated radiotherapy using DIBH three-dimensional conformal radiation therapy (3DCRT), free-breathing (FB)-3DCRT, and FB helical tomotherapy (HT). Dosimetric parameters of cardiac substructures were analyzed, and the results were statistically compared using the Wilcoxon signed-rank test. This study found a significant reduction in the dose to the heart, left anterior descending coronary artery, and ventricles with DIBH-3DCRT and FB-HT compared to FB-3DCRT. While DIBH-3DCRT was very effective in sparing the heart, in some cases, it provided little or no cardiac sparing. FB-HT can be an interesting treatment modality to reduce the dose to major coronary vessels and ventricles and may be of interest for patients with cardiovascular risks who do not benefit from or cannot perform DIBH. These findings highlight the importance of cardiac-sparing techniques for precise delivery of radiation therapy.

## 1. Introduction

Radiotherapy for breast cancer is associated with adverse cardiac events, and patients irradiated for left-sided breast cancer (LSBC) have a higher risk of developing cardiac complications than those irradiated for right-sided breast cancer [1]. Radiation-induced cardiopathy represents a spectrum of early and late effects, including myocardial disease, pericarditis, coronary artery disease, valvular heart disease, and conduction system dysfunction [2]. Radiation damage to the heart is associated with the heart-absorbed dose and irradiated volume. Darby et al. showed that each one-Gray (Gy) dose absorbed by the heart increased the risk of cardiac disease by 7%. This increase has no minimum dose threshold and is independent of the presence of preexisting cardiac risk factors [3]. It is therefore important to reduce the exposure of organs at risk (OARs) to decrease the incidence rate of side effects and improve survival.

Consequently, radiotherapy techniques have evolved to reduce heart exposure to radiation doses without compromising local tumor control or patient survival. The mean heart dose (MHD) of left tangential radiotherapy decreased from 13.3 Gy in the 1970s to 4.7 Gy in the 1990s and to 2.3 Gy in 2006 [4]. Other techniques have been developed, such as deep-inspiration breath-hold (DIBH) approaches, intensity-modulated radiation therapy (IMRT), volumetric-modulated arc therapy (VMAT), tomotherapy, and proton therapy [5].

DIBH three-dimensional conformal radiation therapy (3DCRT) is an irradiation technique in which patients take a deep breath before radiation is delivered and hold their breath during treatment. This approach reduces the heart dose by increasing the distance between the heart and chest wall when the lungs are filled with air. DIBH reduced the MHD by up to 3.4 Gy when compared to a free-breathing approach [6,7]. The clinical use of DIBH with surface-guided radiation therapy (SGRT) provides real-time motion monitoring of the patient’s 3D surface throughout the whole treatment fraction [8].

Sometimes patients cannot perform the DIBH technique, either because they cannot hold their breath or because they cannot reach the gating amplitude. In this case, they are treated with free-breathing (FB)-3DCRT, which increases the risk of a higher dose to the heart, especially to the left anterior descending artery (LADA) [9].

The advancement of software and hardware technology has revolutionized radiation therapy planning, enabling the implementation of IMRT that can be used as a cardiac-sparing technique [10]. Helical tomotherapy (HT), which is a specialized form of IMRT, delivers treatment with a rotating, intensity-modulated fan beam. Delivery is performed while the beam rotates around the patient, and the couch translates through the gantry ring that creates a helical irradiation of the patient. These sophisticated techniques provide treatment planners with the ability to achieve superior distribution of radiation doses within the target volume while minimizing the potential for high-dose exposure to critical OARs. This high level of precision allows for a more personalized approach to radiation therapy, thereby enhancing treatment effectiveness and reducing the risk of harm to surrounding healthy tissues [11,12].

The aim of this study was to compare the dosimetric performances of FB-HT planning with DIBH-3DCRT and FB-3DCRT planning at the cardiac substructure level in patients with early stage LSBC.

To the best of our knowledge, this study is the first to compare these three techniques at the cardiac substructure dosimetric level. While certain cardiac-sparing techniques, such as proton therapy, have shown effectiveness, their limited accessibility prevents them from being considered routine cardiac protection techniques [13]. However, the three techniques studied in our research are commonly employed in breast cancer treatment. As the field progresses with the advent of automatic delineation software, we are moving towards greater precision in dose delivery in radiation therapy.

MHD may not be the most suitable dosimetric parameter for assessing modern radiotherapy techniques that exhibit significantly different dose distributions compared to conventional techniques [14]. The consideration of cardiac substructure dosimetry holds significant potential for improving treatment planning and optimizing radiation delivery to minimize radiation-induced heart disease (RIHD), even for substructures on the right side [15].

## 2. Materials and Methods

### 2.1. Patient Selection

This retrospective single-center dosimetric study was conducted in the Department of Radiation Oncology (ICANS, Strasbourg, France). The study included 10 patients with stage pT1N0 (*n* = 9) or pT2N0 (*n* = 1) left invasive ductal carcinoma who underwent breast-conserving surgery and received adjuvant radiotherapy using hypofractionated DIBH-3DCRT with the clinical use of SGRT. The median age of the patients was 63 years, ranging from 48 to 80 years. Patients with breast implants and those treated for recurrent or metastatic breast cancers were excluded. All patients underwent FB and DIBH computed tomography (CT) scans without contrast injections in the supine position, with their arms above their head, at slice intervals of 2.5 mm. For all patients, the planned target volumes included the whole left breast, and the target volumes were defined according to the ESTRO guidelines [16]. The prescribed dose for the whole breast was 40.05 Gy delivered in 15 fractions. The average volume of the breast PTV was 921 cc, with a range of 612 cc to 1602 cc. The mean volume of the heart was 565 cc, ranging from 410 cc to 660 cc.

### 2.2. Cardiac Segmentation

Cardiac substructures were contoured on simulation CT scans on the Varian Eclipse treatment planning system (Varian Medical Systems, Palo Alto, CA, USA) according to Feng’s atlas to ensure homogeneity and reproducibility of the cardiac segmentation process [17]. The delineated cardiac substructures were the left atrium, left ventricle (LV), right atrium, right ventricle, left main coronary artery, LADA, left circumflex artery, right coronary artery, pulmonary artery, superior vena cava, and ascending and descending aorta. Since Feng’s atlas does not include a specific description of the pericardium, we employed a geometric concept involving a 2 mm wall of the heart volume to approximate the pericardial structure [18].

Cardiac substructure delineation was validated by an experienced senior radiation oncologist. The dose–volume histograms were retrieved for each patient from the initial DIBH treatment plan and the corresponding simulated FB-HT and FB-3DCRT treatment plans on the Varian Eclipse treatment planning system in both cases.

The following evaluation parameters were used for comparison: mean dose (D_mean_); maximum dose (D_max_) to the heart and its substructures and constraints according to the DEGRO [19] and guidelines adapted for hypofractionated schedule [20]; volume receiving at least 35 Gy (V_35Gy_) to the heart; volume receiving at least 26.5 Gy (V_26.5Gy_); volume receiving at least 35.3 Gy (V_35.3Gy_) to the LADA; and volumes receiving at least 4.4 Gy (V_4.4Gy_) and at least 20.3 Gy (V_20.3Gy_) to the LV. Right breast D_mean_, left lung D_mean_, and volume receiving at least 17 Gy (V_17Gy_) to the left lung were also extracted. To compare the effectiveness of the plans, minimum dose to 98% (D_98%_) and to 2% (D_2%_) of the planning target volume (PTV), D_mean_, D_max_, and volume receiving at least 95% of the prescribed dose (V_95%_) to the PTV were evaluated.

The same dose constraints to the OARS were used in the three dose planning calculations: heart V_35Gy_ < 5%, V_17Gy_ < 10%; left lung V_17Gy_ < 30%, D_mean_ < 17 Gy; and dose objective to spinal cord D_max_ < 16 Gy.

### 2.3. DIBH-3DCRT and FB-3DCRT Treatment Planning

For the DIBH-3DCRT plan, using a DIBH CT scan performed with the AlignRT^®^ SGRT system (VisionRT, London, UK), treatment plans were produced by radiation therapists in accordance with national guidelines based on an in-house protocol. The plan was optimized for target coverage with a minimum of 95% of the prescribed dose of 40.05 Gy in 15 fractions to the PTV. To achieve dose homogeneity, 6 MV opposing mono-isocentric tangential conformal photon beams with low-energy 6 or 15 MV segments were used.

The treatment plans were calculated with the Acuros XB dose-to-medium algorithm in the Varian Eclipse treatment planning system (v 15.6). The treatment machine used for modelling was a Clinac Linear Accelerator (Varian, Palo Alto, CA, USA).

### 2.4. FB-HT Treatment Planning

FB-HT plans were calculated from the FB CT scan and optimized with the collapsed cone superposition convolution algorithm (version 2.0.1.1) using an Accuray Radixact tomotherapy treatment machine for modelling. To achieve the same PTV coverage (D98% [Gy]) and heart sparing as the original DIBH-3DCRT plan, the operator set the following three major parameters: a field width of 5 cm, a pitch of 0.287, and a modulation factor of 3. To decrease the radiation dose to the lungs and heart, directional blocking was applied to the parts of the lungs, heart, or liver that were above 5 cm from the PTV. This technique avoids the beam crossing all these organs before arriving in the target, and helps to limit the dose delivered to them.

### 2.5. Statistics

Dose and volume differences between treatment plans were evaluated by the Wilcoxon signed-rank test. Data analyses were conducted using R version 4.2.1 software “www.r-project.org (accessed on 14 August 2022)”. *p*  <  0.05 was considered statistically significant. We employed a collective case study methodology to analyze the data of the patients [21].

## 3. Results

The results for the doses to the OARs and PTV are summarized in Table 1. The mean doses to the selected cardiac substructures are shown in Figure 1. The dose distribution in the target volumes (PTV coverage) indicated that the three dosimetric plans were comparable.

### 3.1. DIBH-3DCRT vs. FB-3DCRT

#### 3.1.1. Overall Outcome

For the 3DCRT plans, the DIBH position achieved significant dose reductions in all evaluated dosimetric parameters of the heart, LADA, pericardium, and ventricles, and D_max_ to the pulmonary artery compared with the FB position (*p*  <  0.05).

#### 3.1.2. Heart Dosimetry

Using DIBH-3DCRT, the heart dosimetric parameters were significantly reduced by 54.2% for the D_mean_, from 2.1 Gy to 1.0 Gy (*p* = 0.02); by 45.2% for the D_max_, from 37.1 Gy to 20.3 Gy (*p* = 0.01); and by 95.4% for V_35Gy_, from 4.3% to 0.2% (*p* = 0.03).

#### 3.1.3. Cardiac Substructures Dosimetry

For the LADA, with DIBH-3DCRT, the dose reductions were 68% for D_mean_, from 13.3 Gy to 4.3 Gy (*p* = 0.04); 51.9% for D_max_, from 32.0 Gy to 15.4 Gy (*p* < 0.01); from 27.3% to 4.9% (*p* = 0.02) for V_26.5Gy_; and from 22.2% to 3.7% (*p* = 0.01) for V_35.3Gy_.

For the pericardium, the dose reductions were 58.9% for the D_mean_, from 3.6 Gy to 1.5 Gy (*p* < 0.01); and 45% for D_max_, from 36.9 Gy to 20.3 Gy (*p* = 0.01).

For the LV, the dose reductions were 62.1% for the D_mean_, from 2.8 Gy to 1.1 Gy (*p* < 0.01); 55.6% for the D_max_, from 35.5 Gy to 15.8 Gy (*p* < 0.01); from 7.6% to 1.1% for V_4.4Gy_ (*p* < 0.01); and from 3.8% to 0.3% (*p* < 0.01) for V_20.3Gy_.

For the right ventricle, the dose reductions were 68.9% for D_mean_, from 3.3 Gy to 1.0 Gy (*p* < 0.01); and 65.5% for D_max_, from 34.7 Gy to 12.0 Gy (*p* < 0.01).

For the pulmonary artery, the dose reductions were 55.2% for D_max_, from 5.3 Gy to 2.4 Gy (*p* = 0.01).

#### 3.1.4. Contralateral Breast and Ipsilateral Lung Dosimetry

For the contralateral breast and ipsilateral lung, the DIBH-3DCRT and FB-3DCRT plans had no significant differences.

### 3.2. DIBH-3DCRT vs. FB-HT

#### 3.2.1. Heart Dosimetry

Compared to the FB-HT setup, the DIBH position was associated with a significant reduction in MHD by 0.6 Gy, from 1.6 Gy to 1.0 Gy (*p* < 0.01); however, a significantly higher D_max_ and V_35Gy_ were observed, with increases from 16.6 Gy to 20.3 Gy (*p* = 0.74) and from 0.0% to 0.2% (*p* = 0.07), respectively.

#### 3.2.2. Cardiac Substructures Dosimetry

Compared to the FB-HT plan, the DIBH plan had significant dose reductions for all parameters of all substructures, except for the parameters of the LADA, D_max_ of the pericardium, and D_max_, V_4.4Gy_, and V_20.3Gy_ of LV.

#### 3.2.3. Contralateral Breast and Ipsilateral Lung Dosimetry

The D_mean_ of the contralateral breast increased by 3.3 Gy with the FB-HT plan. However, in comparison to the DIBH-3DCRT plan, both the D_mean_ and V_17Gy_ of the left lung were significantly reduced with the FB-HT plan. The D_mean_ of the left lung decreased by 33.3%, from 4.8 Gy to 3.2 Gy (*p* < 0.01), and the V*_17Gy_* decreased from 9.9% to 1.4% (*p* < 0.01).

### 3.3. FB-HT vs. FB-3DCRT

#### 3.3.1. Heart Dosimetry

In comparison to FB-3DCRT, FB-HT demonstrated a significant reduction in both D_max_ to the heart and V_35Gy_. The D_max_ was reduced by 55.3%, from 37.1 Gy to 16.6 Gy (*p* < 0.01), while the V_35Gy_ decreased from 4.3% to 0.0% (*p* < 0.01).

#### 3.3.2. Cardiac Substructures Dosimetry

For LADA, the D_mean_ and D_max_ were also reduced with FB-HT plans, by 71.9%, from 13.3 Gy to 3.7 Gy (*p* = 0.04), and by 69.1%, from 32.0 Gy to 9.9 Gy (*p* < 0.01), respectively; furthermore, V_26.5Gy_ and V_35.3Gy_ were reduced from 27.3% to 0.0% and from 22.2% to 0.0%, respectively.

For the LV, all dosimetry parameters were lower with FB-HT plans; the D_mean_ was reduced by 42.9%, from 2.8 Gy to 1.6 Gy, and D_max_ was reduced by 69.7%, from 35.5 Gy to 10.7 Gy (*p* < 0.01). V_4.4Gy_ and V_20.3Gy_ also decreased, from 7.6% to 0.7% (*p* < 0.01) and from 3.8% to 0.0% (*p* < 0.01), respectively.

The D_max_ to the following cardiac substructures was also reduced with the FB-HT plan: pulmonary artery, from 5.3 Gy to 4.4 Gy (*p* < 0.01); pericardium, from 36.9 Gy to 16.5 Gy (*p* = 0.01); and right ventricle, from 34.7 Gy to 13.6 Gy (*p* < 0.01). On the other hand, FB-3DCRT significantly reduced the dose delivered to the aorta, circumflex artery, main coronary artery, right coronary artery, atrium, superior vena cava, and right breast.

#### 3.3.3. Contralateral Breast and Ipsilateral Lung Dosimetry

Compared to FB-3DCRT, FB-HT significantly increased the right breast D_mean_ from 0.3 Gy to 3.6 Gy (*p* < 0.001). However, FB-HT reduced the D_mean_ to the left lung, from 5.1 Gy to 3.2 Gy (*p* < 0.01), and V_17Gy_ from 10.8% to 1.4% (*p* < 0.01).

A comparison of the color-wash dose distribution of all different whole-breast optimization techniques for a typical left breast case is presented in Figure 2.

## 4. Discussion

Breast cancer radiotherapy, especially for left-sided breast cancer, is associated with late cardiovascular complications. RIHD is characterized by acute and chronic changes in the myocardium, pericardium, coronary vasculature, valvular apparatus, and conduction system. Due to the proximity of the target to the heart, especially the LADA, ischemic events are a concern with left-sided breast irradiation, which leads to a 4–16% increase in the risk for major coronary events per Gy of mean MHD [3,22,23]. Thus, cardiac-sparing techniques that decrease heart and coronary artery doses, particularly in the left anterior territory, are critical to minimizing the risk of radiation-induced cardiac dysfunction. In this study, we compared the dose distribution to cardiac substructures of different whole-breast optimization schemes, including FB-3DCRT, DIBH-3DCRT, and FB-HT.

### 4.1. DIBH Effectiveness

In the current study, compared to FB-3DCRT, the DIBH-3DCRT technique allowed for the delivery of lower mean heart and LADA doses, with reductions of 1.2 Gy (equivalent dose in 2 Gy per fraction (EQD2) 0.7 Gy_EQD2_) and 9.6 Gy (7.0 Gy_EQD2_), respectively. This is in accordance with a previously published study that included 25 patients with LSBC treated with adjuvant breast radiotherapy using 50.4 Gy in 1.8 Gy per fraction. DIBH resulted in significant reductions in mean heart and LADA doses of 2.0 Gy (1.2 Gy EQD2) and 10.3 Gy (7.0 GyEQD2), respectively, compared to FB treatments [24]. In the present study, the DIBH-3DCRT technique also delivered lower doses to the pericardium and to both ventricles and reduced D_max_ to the right atrium and pulmonary artery. Consequently, in terms of the general population, the DIBH approach appears to be an effective cardiac-sparing technique for whole-breast radiotherapy, especially for LSBC. Using the Van den Bogaard et al. prediction model, in this study, the 0.7 Gy_EQD2_-MHD reduction with DIBH-3DCRT compared to FB-3DCRT corresponded to a 20% relative risk reduction for acute cardiac events in the first 9 years after treatment.

### 4.2. DIBH Limitations

DIBH has some limitations. First, it takes longer to perform than a standard FB technique. Second, DIBH can be sometimes difficult or impossible for patients to perform, especially older patients, either because they are in a difficult-to-reach position or because they struggle to hold their breaths for 15–25 s [25].

Accordingly, not every patient will benefit from DIBH. Rochet et al. evaluated a cohort of 35 LSBC patients and reported only 75% had dosimetric benefits from DIBH, meaning a heart D_mean_ reduction ≥ 0.9 Gy [26]. This is in line with our results. Two patients had no dosimetric benefits from DIBH, with no reduction of the D_max_ to the heart, and two patients had only a modest D_mean_ reduction (<0.15 Gy). For the LADA, the D_max_ was not reduced for two patients, and even with the DIBH technique, these patients received a maximum dose of 39.0 and 40.4 Gy to the LADA.

Due to the intricacies involved in this procedure, it is crucial to carefully select patients who are likely to gain the most advantage from it. Several studies have highlighted the predictive value of anatomical and volume parameters in determining heart exposure, including factors such as breast size, chest wall separation, maximum heart depth, tumor bed location, and heart volume within the radiation field [27,28].

Corradini et al. estimated that the cardiac risk reduction from DIBH was more prominent among older patients (>70 years) due to a higher frequency of worse baseline cardiac risk factors [29]. Unfortunately, since this technique requires good collaboration, good pulmonary function, and anxiety management, the population most likely to benefit from it is also the one least likely to be considered suitable for it [30]. If patients cannot meet the stated requirements, they can receive FB-3DCRT, which has an increased risk of higher doses to the heart and cardiac substructures.

### 4.3. FB-HT as an Alternative

Mathieu et al. found that HT can be a cardiac-sparing alternative to DIBH, with a significant MHD reduction compared to FB-3DCRT [31]. In the current study, compared to FB-3DCRT, the FB-HT plan did not significantly reduce MHD. However, it significantly reduced the D_max_ and V_35Gy_ to the heart and the dosimetric parameters measured for LADA and LV. For patients who had no dosimetric benefits from DIBH, FB-HT reduced D_max_ to the LADA and LV. Since these cardiac substructures are correlated with cardiac events, patients who had no dosimetric benefit from DIBH but required cardiac protection may benefit from FB-HT [23,32,33]. Compared to FB-3DCRT, FB-HT also significantly reduced LV-V_5Gy EQD2,_ which appeared to be the most important prognostic dose–volume parameter [23]. The reduction in D_max_ to the heart, LV, and LADA could have clinical benefits, as suggested by previous studies that found subclinical changes detected with early imaging after a high dose to the myocardium [34,35,36,37,38]. Evans et al. found that the amount of heart tissue exposed to 20 Gy correlated with increased focal cardiac FDG uptake [39]. However, controversial results have been reported regarding the clinical outcomes predicted by subclinical myocardial damage [40,41,42].

### 4.4. Mean Heart Dose and Cardiac Substructures

To our knowledge, our study has evaluated the most cardiac substructures in the dose distribution of FB-HT, FB-3DCRT, and DIBH-3DCRT plans. Cardiac substructures are not delineated in daily practice, mainly because MHD is widely considered the only dosimetric parameter to optimize. Recent studies have demonstrated that irradiation to specific cardiac substructures was significantly associated with distinctive cardiac adverse events [43]. In line with the previous literature, the results of this study confirmed that MHD is not the best relevant parameter for a full understanding of left-sided breast dosimetry, and MHD is insufficient to predict the individual patient dose to the LV and coronary arteries, especially the LADA [44,45,46]. With the development of software for the autosegmentation of cardiac structures, it is highly expected that cardiac substructure delineation could be routinely performed in the near future [47,48].

### 4.5. FB-HT Limitations

Although FB-HT allowed for decreases in the dosimetric parameters predictive of coronary events, these reductions occurred at the cost of a higher D_mean_ and D_max_ dose to other cardiac substructures, especially the base of the heart, and higher D_mean_ to the contralateral breast. The potential detrimental long-term effects of the low radiation doses delivered, especially to the contralateral breast, are a cause of general concern [49]. However, no clinical studies to date have shown as high a risk as the theoretical models predicted [50]. 

It is challenging to know the clinical impact of this dose distribution rather than that with a higher dose in the apex. The answer may depend on the clinical characteristics of the patients, such as age or the presence of cardiovascular risk factors. There is no clear consensus regarding the acceptable low doses to the base of the heart, since dosimetric optimization is routinely performed on the overall cardiac dose. The DEGRO breast cancer expert panel recommends constraints to the heart substructures, mentioning only the LADA and LV [14]. Dosimetric parameters can be individualized to patients, and concomitant cardiovascular risk factors may represent greater competitive risk factors for death than secondary malignancy [41]. Another limitation of FB-HT is the need for a longer among of time for treatment design because the parameters are adjusted and optimized repeatedly.

### 4.6. What Technique Should Be Used?

DIBH produced a better cardiac-sparing dosimetric plan; however, the accumulation of the literature about the importance of heart protection and substructure consideration should allow other techniques to be used when DIBH is not suitable. FB-HT reduced the dose to cardiac substructures that are correlated with cardiac events and could therefore be a cardiac-sparing alternative to DIBH for three main reasons: (1) not every patient will benefit from DIBH, (2) the population most likely to benefit from DIBH is also the one least likely to be considered suitable for it, and (3) FB-HT produced a dosimetric benefit when DIBH did not.

Our study has certain limitations that should be considered. Firstly, due to the very small number of cases studied [10], the generalizability of the findings to a larger population may be limited. Secondly, we did not specifically address the issue of patient selection for the DIBH technique. Implementing DIBH can be challenging and may not be suitable for all patients. It requires patient training, and only those who can comfortably hold their breath for an appropriate duration can be selected for this technique. Therefore, while our study provides valuable insights, its findings may not be directly applicable to all patients undergoing radiation therapy for LSBC.

## 5. Conclusions

Different cardiac-sparing optimization schemes are possible when treating LSBC. While DIBH-3DCRT appears to be a very effective cardiac-sparing technique, tomotherapy can be an interesting treatment modality to reduce the doses to major coronary vessels and ventricles, which may be of interest for patients with cardiovascular risks who are unable to benefit from DIBH.

## Figures and Tables

**Figure 1 cancers-15-03406-f001:**
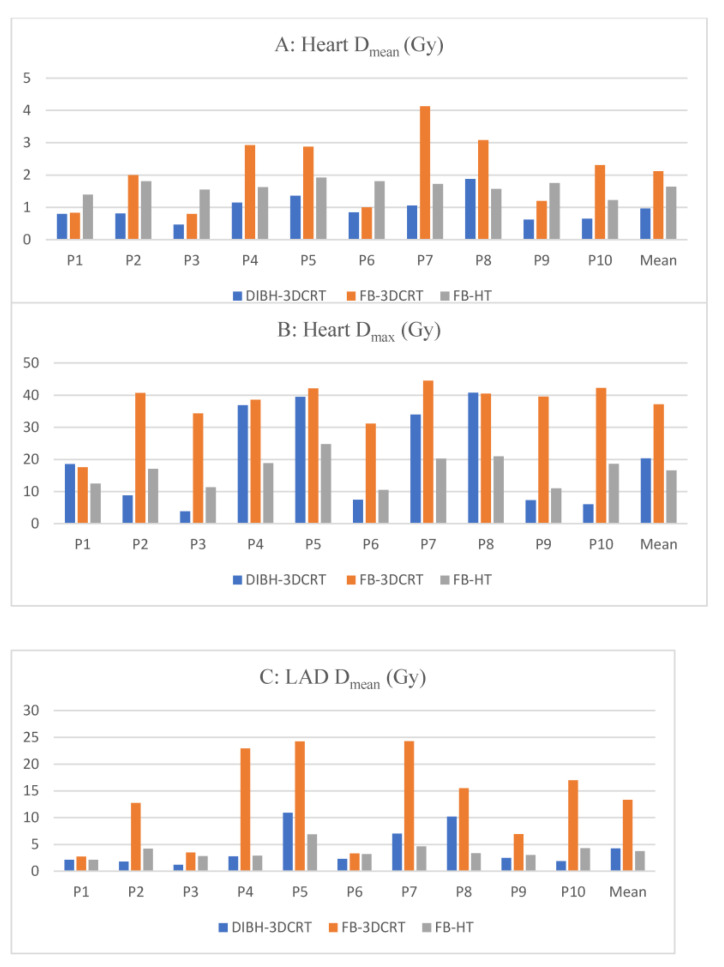
Dosimetric comparisons between heart and cardiac substructures: (**A**) mean heart dose [Gy], (**B**) maximum heart dose [Gy], (**C**) mean left anterior descending artery dose [Gy], (**D**) maximum left anterior descending artery dose [Gy], (**E**) mean left ventricle dose [Gy], and (**F**) maximum left ventricle dose [Gy]. Blue bars: DIBH-3DCRT; orange bars: FB-3DRCT; and gray bars: FB-HT. Abbreviations: DIBH-3DCRT: three-dimensional conformal radiation therapy with deep-inspiration breath-hold; D_max_: maximum dose; D_mean_: mean dose; FB-3DCRT: free-breathing three-dimensional conformal radiotherapy; FB-HT: free-breathing helical tomotherapy; Gy: Gray; LADA: left anterior descending artery; and LV: left ventricle.

**Figure 2 cancers-15-03406-f002:**
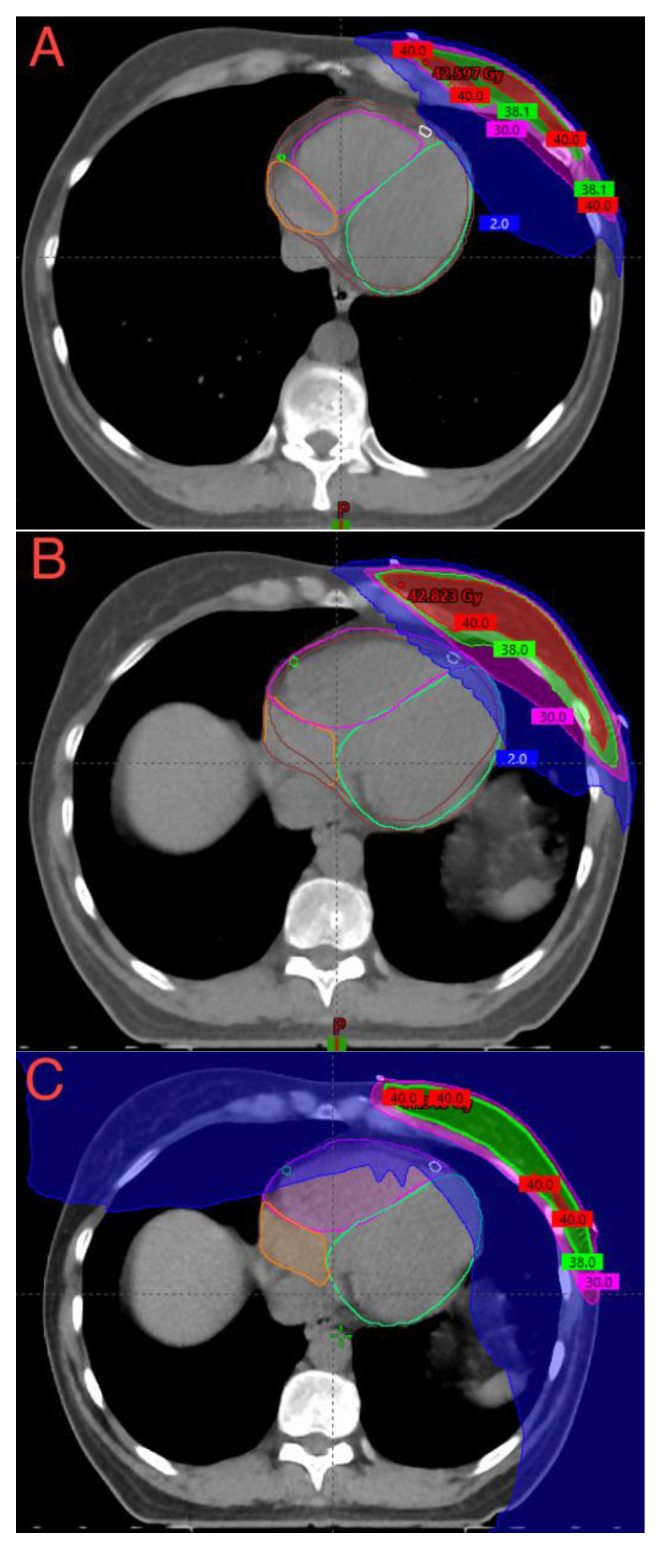
Color-wash dose distributions of (**A**) DIBH-3DCRT plan; (**B**) FB-3DCRT plan; and (**C**) FB-HT plan (prescribed dose: 40.05 Gy/15 fx). Isodose level: blue, 2.00 Gy; pink, 30.00 Gy; green 38.05 Gy; and red 40.05 Gy. Cardiac substructures: white, LADA; light green, left ventricle; pink, right ventricle; green, right coronary artery; and orange, right atrium. Abbreviations: DIBH-3DCRT: deep-inspiration breath-hold three-dimensional conformal; FB-3DCRT: free-breathing three-dimensional conformal radiotherapy; FB-HT: free-breathing helical tomotherapy; fx: fractions; Gy: Gray; and LADA: left anterior descending artery.

**Table 1 cancers-15-03406-t001:** Comparison of dosimetry metrics for all 10 patients (Mean  ±  SD).

Volume	Dosimetric Parameter	DIBH-3DCRT	FB-3DCRT	FB-HT	*p*-Value		
					DIBH-3DCRT vs. FB-3DCRT	DIBH-3DCRT vs. FB-HT	FB-3DCRT vs. FB-HT
PTV	D98% (Gy)	37.4 ± 0.4	37.5 ± 0.9	37.5 ± 0.5	0.97	0.48	0.68
	V38.1 Gy (%)	96.2 ± 1.0	95.5 ± 3.3	96.9 ± 1.1	0.97	0.27	0.27
	D2% (Gy)	42.5 ± 0.3	42.6 ± 1.1	40.9 ± 0.3	0.12	1.08	1.08
	Dmax (Gy)	43.6 ± 0.3	43.9 ± 1.6	42.7 ± 0.6	0.43	**<0.01**	**<0.01**
	Dmean (Gy)	40.3 ± 0.3	40.3 ± 0.6	39.7 ± 0.2	0.54	**<0.01**	**<0.01**
Heart	Dmean (Gy)	1.0 ± 0.4	2.1 ± 1.1	1.6 ± 0.2	**<0.01**	**<0.01**	0.47
	Dmax (Gy)	20.3 ± 15.6	37.1 ± 7.9	16.6 ± 5.0	**<0.01**	0.74	**<0.01**
	V35.0 Gy (%)	0.2 ± 0.5	4.3 ± 9.2	0.0 ± 0.0	**0.03**	0.07	**<0.01**
Ascending aorta	Dmean (Gy)	0.5 ± 0.1	0.6 ± 0.1	2.1 ± 1.1	0.32	**<0.01**	**<0.01**
	Dmax (Gy)	1.0 ± 0.2	1.1 ± 0.2	7.4 ± 4.3	**0.05**	**<0.01**	**<0.01**
Descending aorta	Dmean (Gy)	0.3 ± 0.1	0.3 ± 0.1	0.9 ± 0.2	0.26	**<0.01**	**<0.01**
	Dmax (Gy)	0.5 ± 0.2	0.6 ± 0.1	1.8 ± 0.5	0.36	**<0.01**	**<0.01**
Circumflex artery	Dmean (Gy)	0.7 ± 0.2	0.8 ± 0.2	1.3 ± 0.3	0.08	**<0.01**	**<0.01**
	Dmax (Gy)	0.8 ± 0.2	0.9 ± 0.3	1.7 ± 0.4	0.14	**<0.01**	**<0.01**
Main coronary artery	Dmean (Gy)	0.5 ± 0.1	0.7 ± 0.2	1.2 ± 0.2	0.08	**<0.01**	**<0.01**
	Dmax (Gy)	0.7 ± 0.2	0.9 ± 0.2	1.5 ± 0.3	0.06	**<0.01**	**<0.01**
Right coronary artery	Dmean (Gy)	0.6 ± 0.2	0.7 ± 0.2	2.3 ± 0.6	0.1	**<0.01**	**<0.01**
	Dmax (Gy)	0.8 ± 0.2	1.0 ± 0.3	4.8 ± 2.5	0.12	**<0.01**	**<0.01**
LADA	Dmean (Gy)	4.3 ± 3.7	13.3 ± 8.8	3.7 ± 1.3	**0.04**	0.23	**<0.01**
	Dmax (Gy)	15.4 ± 15.7	32.0 ± 13.5	9.9 ± 5.7	**<0.01**	0.91	**<0.01**
	V26.5 Gy (%)	4.9 ± 9.5	27.3 ± 24.4	0.0 ± 0.0	**0.02**	0.08	**<0.01**
	V35.3 Gy (%)	3.7 ± 7.9	22.2 ± 20.9	0.0 ± 0.0	**<0.01**	0.17	**<0.01**
Pulmonary artery	Dmean (Gy)	0.8 ± 0.1	0.9 ± 0.3	1.6 ± 0.4	0.34	**<0.01**	**<0.01**
	Dmax (Gy)	2.4 ± 0.5	5.3 ± 6.0	4.4 ± 2.1	**<0.01**	**<0.01**	**<0.01**
Left atrium	Dmean (Gy)	0.4 ± 0.1	0.5 ± 0.1	0.9 ± 0.01	0.27	**<0.01**	**<0.01**
	Dmax (Gy)	1.1 ± 0.3	1.6 ± 1.1	2.3 ± 0.7	0.38	**<0.01**	**<0.01**
Right atrium	Dmean (Gy)	0.3 ± 0.1	0.4 ± 0.1	1.3 ± 0.3	0.11	**<0.01**	**<0.01**
	Dmax (Gy)	0.7 ± 0.2	1.0 ± 0.4	5.6 ± 2.2	**0.02**	**<0.01**	**<0.01**
Pericardium	Dmean (Gy)	1.5 ± 0.9	3.6 ± 2.0	2.2 ± 0.3	**<0.01**	0.03	0.2
	Dmax (Gy)	20.3 ± 15.6	36.9 ± 8.3	16.5 ± 4.9	**0.01**	0.85	**<0.01**
Superior vena cava	Dmean (Gy)	0.4 ± 0.1	0.4 ± 0.1	2.5 ± 1.7	0.73	**<0.01**	**<0.01**
	Dmax (Gy)	0.5 ± 0.1	0.6 ± 0.2	5.0 ± 2.8	0.65	**<0.01**	**<0.01**
Left ventricle	Dmean (Gy)	1.1 ± 0.4	2.8 ± 1.5	1.6 ± 0.2	**<0.01**	**<0.01**	0.21
	Dmax (Gy)	15.8 ± 15.4	35.5 ± 10.2	10.8 ± 5.0	**<0.01**	0.74	**<0.01**
	V4.4 Gy (%)	1.1 ± 1.6	7.6 ± 5.2	0.7 ± 0.6	**<0.01**	0.42	**<0.01**
	V20.3 Gy (%)	0.3 ± 0.5	3.8 ± 3.3	0.0 ± 0.0	**<0.01**	0.08	**<0.01**
Right ventricle	Dmean (Gy)	1.0 ± 0.6	3.3 ± 2.5	2.1 ± 0.4	**<0.01**	**<0.01**	0.53
	Dmax (Gy)	12.0 ± 15.3	34.7 ± 11.1	13.6 ± 4.5	**<0.01**	0.09	**<0.01**
Right breast	Dmean (Gy)	0.3 ± 0.1	0.3 ± 0.1	3.6 ± 0.5	0.62	**<0.01**	**<0.01**
Left lung	Dmean (Gy)	4.8 ± 0.7	5.1 ± 0.9	3.2 ± 0.6	0.31	**<0.01**	**<0.01**
	V15.0 Gy (%)	9.9 ± 1.9	10.8 ± 2.6	1.4 ± 0.8	0.44	**<0.01**	**<0.01**

Abbreviations: DIBH-3DCRT: deep-inspiration breath-hold three-dimensional conformal; Dmax: maximum dose; Dmean: mean dose; Dx%: minimum dose to x% of the volume; FB-3DCRT: free-breathing three-dimensional conformal radiotherapy; FB-HT: free-breathing helical tomotherapy; Gy: Gray; LADA: left anterior descending artery; PTV: planning target volume; SD: standard deviation; and VxGy: volume receiving at least x Gy.

## Data Availability

The data presented in this study are available on request from the corresponding author.

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
