# Peer review of "Evaluation of Cardiac Substructures Exposure of DIBH-3DCRT, FB-HT, and FB-3DCRT in Hypofractionated Radiotherapy for Left-Sided Breast Cancer after Breast-Conserving Surgery: An In Silico Planning Study"

_cancers, 2023, doi:10.3390/cancers15133406_

Round 1

Reviewer 1 Report

This study is about planning approaches with different techniques of 3DCRT.

In the era of novel techniques like proton therapy I guess this has no clinical relevance

Reviewer 2 Report

1) From line 75 to 81, you switched from IMRT to HT as the common reader know that HT is a form of IMRT. Please, specify it for non-insiders.

2) Line 87: you wrote "locoregional". I didn't understand if some patients underwent also lymph node irradiation, including the supraclavicular basin. If so, please, clarify how you managed the field junction between the breast target and the lymph node drainages, considering that the first could overlap the second in case of voluntary breathing movement.

3) Line 116: Here I read a "spinal cord" constraint that's obviously not overcome if irradiation was exclusively extended to the residual breast (without lymph node irradiation).

4) Lines 194-195: value numbers should be inverted: from 16.6 to 20.3 and from 0.0 to 0.2%.

5) Line 202: "Dmean increased by 3.3 Gy with the FB-HT plan", specify that this refers to contralateral breast.

6) Lines 207-208 should be rephrased since they are unclear.

7) Lines 251-252: the values reported are inconsistent with those in Table 1, where the differences were 1.1 Gy (not 1.2 Gy) and 9 Gy (not 9.05 Gy). Moreover, the EQD2 values are inconsistent with those calculated by me with an alfa/beta of 10: 1.1 Gy in 15 fractions is equal to 0,92 Gy in EQD2, while 9 Gy in 15 fractions is equal to 7.95 Gy in EQD2. Specify how you calculated these values and which alfa/beta for normal tissues you used, taking in mind that an alfa/beta value of 10 is suitable for cancer cells rather than for normal tissues. Please, re-check all values in the manuscript. All, even those in tables and figures.

8) Lines 255-256: re-check also these values.

9) Line 261: I still didn't understand how you calculated EQD2. Re-check values reported.

10) In section 4.2 DIBH limitations, you could cite PMID: 33788746, where the authors reported an increased radiation exposure of the heart with increasing breast size as this may require a greater tangential field distance crossing a greater portion of the heart. Read and discuss these findings.

11) Lines 293-294: "The reduction in Dmax to the heart, LV and LADA could have clinical benefits, as suggested by previous studies that found subclinical changes detected with early imaging after a high dose to the myocardium (25–29)", I agree with this observation since the LADA may work as a serial organ at risk: a focal damage on it may be as harmful as on the whole organ.

12) I'd appreciate your point of view about the use of VMAT in this scenario, instead of HT. This facility has less spread while VMAT is common among radiotherapy departments. VMAT is a rotational technique like HT and treatment delivery would be more rapid. Please, add your considerations about the expected differences between FB-VMAT and FB-HT.

English is good. Some sentences require revision, as specified above.

Reviewer 3 Report

Thank you for providing the opportunity to review the manuscript, titled, “Evaluation of Cardiac Substructures exposure of DIBH-3DCRT SGRT, FB-HT and FB-3DCRT in Hypofractionated Radiotherapy for Left-Sided Breast Cancers After Breast-Conserving Surgery: An In-Silico Planning Study”. The authors compared three techniques for left side breast cancer patients (DIBH-3DCRT, FB-HT, and FB-3DCRT). I carefully reviewed this; however, it is difficult to accept this manuscript as it stands. One of the main reasons is the lack of clear novelty. As the authors cite, many investigators have conducted planning studies on cardiac dose reduction during left breast cancer irradiation, and we do not see any novelty in this study compared to those studies. Some comments are given below, which may be useful as a follow-up study.

1.      Is the “SGRT” in the title necessary? It does not seem particularly important in the manuscript and is not in the abstract, so we think it is unnecessary.

2.      Please mention the novelty in the Introduction section.

3.      There is too little information about FB-HT, please mention details such as whether it was performed by TomoDirect, how respiratory migration measures were taken, etc.

4.      Table1: FB-3DRT ⇒ FB-3DCRT

5.      Too little information about the patient. Without information on the volume and positioning of the heart, the size of the target, etc., the reader has no way to consider the information. As you also mention, DIBH is not effective for all patients, so I think you should mention what characteristics of the patients it is not effective for.

6.      Figure 2: what do the brown structures in A and B refer to?

7.      Discussion: is Line 268, Breath-hold time, 20-30s too long? In general, I have the impression that one breath hold is about half of that. Please consider citing the literature if using the above value.

8.      Small letters in Line 295: et al.

Reviewer 4 Report

The topic is interesting and relevant. The sample size (only 10 patients) is very small and not sufficient to draw significant and strong conclusions. The text lacks a description of the weaknesses of the study.

The clinical characteristics of the patients, e.g. age of the patients, should be described in the methods, as it may have a significant influence on the feasibility of deep inspiration irradiation, and it may also influence the size of the heart and therefore the anatomical displacement of the heart in deep inspiration. The methods should also describe which fractionation schemes were used for the patients, which area was irradiated (breast only, also regional lymph nodes, etc.) and whether the tumour bed was irradiated at the same time (simultaneous boost).

In the simple summary, there is a statement that IMRT is a cardiac-sparing approach - this argument should be supported by a reference in the introduction or the statement should be retracted.

no major comments

Round 2

Reviewer 2 Report

I would commend the authors for their thorough revision job. Thank you for having addressed all issues arisen by me. The manuscript improved a lot and is now suitable for publication.

Reviewer 3 Report

Appropriate corrections have been made and there are no concerns.

Reviewer 4 Report

I agree with most of the corrections.

 Given that  only 10 patients were included in the study,  it is a case study and it needs to be clearly written in the manuscript.

I would suggest the following corrections: due to a very small number of cases studied (10), the word "relatively" in line 378 is redundant.

The title should be corrected to the singular: Left sided breast cancer (without "s")
